# On Boussinesq’s Problem for a Power-Law Graded Elastic Half-Space on Elliptical and General Contact Domains

**DOI:** 10.3390/ma16124364

**Published:** 2023-06-13

**Authors:** Emanuel Willert

**Affiliations:** Institute of Mechanics, Technische Universität Berlin, Sekr. C8-4, Straße des 17. Juni 135, 10623 Berlin, Germany; e.willert@tu-berlin.de

**Keywords:** functionally graded materials, normal contact, elliptical contacts, almost axisymmetric contacts, Fabrikant’s approximation, power-law indenters

## Abstract

The indentation of a power-law graded elastic half-space by a rigid counter body is considered in the framework of linear elasticity. Poisson’s ratio is assumed to be constant over the half-space. For indenters with an ellipsoidal power-law shape, an exact contact solution is derived, based on the generalizations of Galin’s theorem and Barber’s extremal principle for the inhomogeneous half-space. As a special case, the elliptical Hertzian contact is revisited. Generally, elastic grading with a positive grading exponent reduces the contact eccentricity. Fabrikant’s approximation for the pressure distribution under a flat punch of arbitrary planform is generalized for power-law graded elastic media and compared with rigorous numerical calculations based on the boundary element method (BEM). Very good agreement between the analytical asymptotic solution and the numerical simulation is obtained for the contact stiffness and the contact pressure distribution. A recently published approximate analytic solution for the indentation of a homogeneous half-space by a counter body, whose shape slightly deviates from axial symmetry but is otherwise arbitrary, is generalized for the power-law graded half-space. The approximate procedure for the elliptical Hertzian contact exhibits the same asymptotic behavior as the exact solution. The approximate analytic solution for the indentation by a pyramid with square planform is in very good agreement with a BEM-based numerical solution of the same problem.

## 1. Introduction

Functionally graded materials (FGMs) are widespread amongst tribological systems in biology [1], biotechnology [2], and technics [3]. The main reason for that is their enhanced ability to withstand various forms of mechanical damage associated with contact and deformation [4]. The analysis of mechanical contacts of FGMs started in the framework of geo- and soil mechanics [5], as early as in the 1940s. Only decades later, the tribological perspectives of FGMs became a focus of research [6], which sparked a broad series of works on their contact mechanical properties (for a comprehensive, up-to-date overview, see the handbook [7], p. 251 ff.).

Meanwhile, tribology and materials science of FGMs and functionally graded coatings have developed into large scientific branches within their respective fields, with a very broad corresponding literature (see, e.g., [8,9,10]). However, in the present manuscript, we will concern ourselves only with the contact mechanics of FGMs.

In that regard, several forms of functional material inhomogeneity have been considered in the literature [11]. Nevertheless, only a few cases allow for a closed-form analytical treatment. One of them is constituted by materials, which exhibit power-law grading of the elastic properties in the normal direction, i.e., into the material.

While axisymmetric contacts of power-law graded elastic materials have been studied in depth, and the corresponding state of the start is basically on par with the one for homogeneous media—including general solutions for non-adhesive normal contacts [12,13], adhesive normal contacts in the JKR [14,15] and Dugdale-Maugis descriptions [16,17], as well as tangential contacts with friction [18,19]—the literature about analytic contact mechanics of power-law graded materials on more general contact domains is still very scarce. Nonetheless, a big step forward in this regard was already made by Rostovtsev [20,21] in the 1960s, who generalized Galin’s theorem [22] (p. 120) for this material class. Based on Rostovtsev’s work, much later Argatov [23], in the framework of asymptotic solutions for multiple contacts, provided the exact solution for the elliptical Hertzian contact problem of power-law graded materials.

On the other hand, several exact or approximate analytic techniques have been developed for simply connected contacts of homogenous materials with general contact domains, e.g., Fabrikant’s approximation for the pressure distribution under a flat punch with arbitrary planform [24], or a very recently published closed-form asymptotic solution for almost axisymmetric contact profiles [25]. As it turns out, these techniques can be generalized for power-law graded media, as will be shown in the present manuscript. This will allow for the very effective, analytic treatment of contact problems for FGMs on non-standard contact domains in different fields of tribology or material testing.

The remainder of the manuscript is organized as follows. In Section 2, several theoretical basics are repeated briefly, which are required for the following analysis. Section 3 provides the exact general contact solution for the indentation of a power-law graded elastic half-space by a rigid, ellipsoidal, power-law indenter, and, as an illustrative example, the elliptical Hertzian contact is revisited. The possibility of generalizing Fabrikant’s approximation for the inhomogeneous case is explored in Section 4, based on which, in Section 5, the generalization of the recently published approximate solution for almost axisymmetric contacts of homogeneous media is derived and illustrated by the elliptical Hertzian contact and the indentation by a rigid pyramid with square planform. Some conclusive remarks finish the manuscript.

## 2. Fundamentals

### 2.1. Problem Formulation

Let us consider the non-adhesive, frictionless normal contact of a rigid indenter and an elastic body, which obeys the restrictions of the half-space approximation. For brevity, we will call this class of contact problems the “Boussinesq problem”. The elastic body shall exhibit functional elastic grading in power-law form in the normal direction, i.e., Young’s modulus *E* as a function of depth *z* is defined by
(1)$$E\left(z\right)=E_{0}{\left(\frac{z}{z_{0}}\right)}^{k},\;\;\;\;\;\left|k\right|<1,$$
with some constants *E*_0_ and *z*_0_ (mathematicians might treat *E*_0_/(*z*_0_)*^k^* as one constant, but in the above formulation *E*_0_ and *z*_0_ have immediate physical meaning). Poisson’s ratio *ν* shall be constant. Effects of microscopic surface roughness, inelasticity, or surface tension are neglected. The gap between the contacting bodies at the moment of first contact constitutes the contact profile *f*(*x*,*y*), with the cartesian coordinates *x* and *y* in the contact plane. The generalization for the contact of two elastic bodes is straightforward if both materials have the same exponent *k* of elastic grading. The case of different values for *k* is mathematically very complicated, although, very recently also for this problem, analytic contact solutions have been provided [26].

### 2.2. Fundamental Solution for Point Loading

If a normal point load *F* acts on the surface of an elastic half-space with elastic grading in the form (1), the half-space surface will experience elastic normal displacements [27]:(2)$$\begin{array}{c} w\left(r\right)=\frac{F}{\pi \tilde{c}r^{1+k}},\;\;\;\;\tilde{c}={\left[\left(1+k\right)\Gamma \left(1+\frac{k}{2}\right)\right]}^{2}\frac{E_{0}}{\left(1-\nu^{2}\right)z_{0}^{k}C},\\ \;\;\;C=\beta \sin\left(\frac{\pi \beta }{2}\right)\Gamma \left(\frac{3+k+\beta }{2}\right)\Gamma \left(\frac{3+k-\beta }{2}\right),\;\;\;\;\;\beta =\sqrt{\left(1+k\right)\left(1-\frac{k\nu }{1-\nu }\right)}\;\;, \end{array}$$
with the Gamma function Γ and the polar radius *r* from the force application point.

Hence, for a pressure distribution *p*(*x*,*y*), the resulting surface normal displacement is provided by the convolution
(3)$$w\left(x,y\right)=\frac{1}{\pi \tilde{c}}\iint_{\Omega }\frac{p\left(x^{\prime },y^{\prime }\right)dx^{\prime }dy^{\prime }}{\sqrt{{\left[{\left(x^{\prime }-x\right)}^{2}+{\left(y^{\prime }-y\right)}^{2}\right]}^{1+k}}},$$
with the contact domain Ω.

### 2.3. Generalization of Galin’s Theorem

Rostovtsev [20,21] has generalized Galin’s theorem for the homogeneous elastic half-space, as he proved that a pressure distribution of the form
(4)$$p\left(x,y\right)=S_{m}\left(x,y\right){\left(1-\frac{x^{2}}{a^{2}}-\frac{y^{2}}{b^{2}}\right)}^{\frac{k-1}{2}},\;\;\;\;\frac{x^{2}}{a^{2}}+\frac{y^{2}}{b^{2}}<1,$$
with some polynomial *S_m_*(*x*,*y*) of the order *m*, corresponds to a polynomial normal surface displacement of the same order in the contact domain, and vice versa. Notably, Galin’s theorem describes a logical equivalence, i.e., it is valid in both directions. Rostovtsev proved both directions of the generalization separately in the publications [20,21], respectively.

### 2.4. Generalization of Barber’s Extremal Normal Force Principle

Another important theorem, which will be used in the following, is Barber’s [28] principle for the homogeneous elastic half-space, that the contact domain in the corresponding indentation problem (if the domain is not fixed) can be determined from the condition
(5)$${\frac{\delta F}{\delta \Omega }|}_{d=const.}=0,$$
with the indentation depth *d*, and the normal force integral [29] (p. 52)
(6)$$F\left(\Omega ,d\right)=\iint_{\Omega }\left[d-f\left(x,y\right)\right]\;p^{*}\left(x,y\right)dxdy,$$
with the pressure distribution *p** due to the unit indentation of the elastic half-space by a flat punch with the planform Ω.

Barber’s proof is based on a harmonic function representation of the elastic displacements. Let us briefly consider an alternative proof, which is far easier to generalize mathematically for an inhomogeneous half-space. Equation (6) can be written in the form
(7)F(Ω,d)=K(Ω)d−G(Ω),
with the contact stiffness *K* of the flat punch contact and a functional *G* of the indenter profile,
(8)$$\begin{array}{c} K\left(\Omega \right)=\iint_{\Omega }p^{*}\left(x,y\right)dxdy,\\ G\left(\Omega \right)=\iint_{\Omega }f\left(x,y\right)\;p^{*}\left(x,y\right)dxdy. \end{array}$$

As was first pointed out by Mossakovskii [30] in the context of axisymmetric normal contacts without a slip of homogeneous elastic materials, the incremental difference between two contact configurations with the contact domains Ω and Ω + dΩ (dΩ is the change in the contact domain due to the change in indentation depth from *d* to *d* + d*d*) can be thought of as an infinitesimal indentation d*d* by a rigid flat punch with the planform Ω. Therefore, the (incremental) contact stiffness *K*(Ω) applies to any (frictionless) normal contact with the same contact domain, and we have from Equation (7)
(9)$$\frac{dF}{dd}=K=\frac{\partial F}{\partial d}+\frac{\delta F}{\delta \Omega }\frac{d\Omega }{dd}=K+\frac{\delta F}{\delta \Omega }\frac{d\Omega }{dd},$$
from which Equation (5) can be immediately deduced. Moreover, the flat punch superposition idea is in no way bound to the homogeneity of the elastic half-space. Hence, Barber’s principle can be also applied straightforwardly for the power-law graded half-space.

### 2.5. Elliptical Flat Punch

The pressure distribution under an elliptical rigid flat punch with the half-axes *a* and *b* is an indispensable basis for the analysis in the following sections. It follows immediately as a special case of Rostovtsev’s (or Galin’s) theorem, because, according to Equation (4), a pressure distribution
(10)$$p\left(x,y\right)=p_{0}{\left(1-\frac{x^{2}}{a^{2}}-\frac{y^{2}}{b^{2}}\right)}^{\frac{k-1}{2}},\;\;\;\;\frac{x^{2}}{a^{2}}+\frac{y^{2}}{b^{2}}<1,$$
will produce a constant surface normal displacement inside the elliptical contact domain. For convenience, we may rewrite the pressure distribution in polar coordinates {*r*,*φ*} as
(11)$$p\left(r,\varphi \right)=p_{0}{\left(1-\frac{r^{2}}{R{\left(\varphi \right)}^{2}}\right)}^{\frac{k-1}{2}},\;\;\;\;r<R\left(\varphi \right),$$
with the contour of the contact domain (without loss of generality, we put *a* > *b*)
(12)$$R\left(\varphi \right)=b{\left(1-e^{2}\cos^{2}\varphi \right)}^{-1/2},\;\;\;e^{2}=1-\frac{b^{2}}{a^{2}}.$$
The total normal force is defined by
(13)$$F=\int_{0}^{2\pi }\int_{0}^{R\left(\varphi \right)}p\left(r,\varphi \right)\;rdrd\varphi =\frac{2\pi }{1+k}p_{0}ba,$$
and the indentation depth follows from Equation (3) as
(14)$$\begin{array}{l} d=w\left(x=0,y=0\right)=\frac{1}{\pi \tilde{c}}\int_{0}^{2\pi }\int_{0}^{R\left(\varphi \right)}\frac{p\left(r,\varphi \right)}{r^{k}}\;drd\varphi =\frac{2p_{0}b^{1-k}}{c}K_{k}\left(e\right),\\ c=\tilde{c}\cos\left(\frac{k\pi }{2}\right),\;\;\;\;K_{k}\left(e\right)=\int_{0}^{\pi /2}{\left(1-e^{2}\cos^{2}\varphi \right)}^{\frac{k-1}{2}}d\varphi . \end{array}$$

## 3. Contact with a Power-Law Ellipsoidal Indenter

An interesting contact problem, which always exhibits an elliptical contact domain because of Galin’s theorem, is the indentation by a power-law ellipsoidal indenter, with a constant eccentricity of the elliptical horizontal indenter cross sections. The contact profile reads
(15)$$f\left(r,\varphi \right)=B_{n}r^{2n}{\left(1-\varepsilon^{2}\cos^{2}\varphi \right)}^{n},\;\;\;\;\;n\in N^{+},$$
with the indenter eccentricity *ε* and some constant *B_n_*. A very common special case of such a profile is the elliptical Hertzian contact with *n* = 1. For the contact with a homogeneous elastic half-space, very recently a general exact solution has been provided by the author for the Boussinesq problem with this class of counter bodies [31]—note that the case *n* = 2 was already considered for homogeneous materials by Shtayerman [32] (pp. 257 ff). Based on the fundamental relations laid out in the previous section, the general homogeneous solution can be generalized for the contact with a power-law graded elastic half-space without severe difficulties.

The following procedure is also applicable to more general elliptical contact profiles if the profile can be expanded as a series of the power-law functions (15). However, if the series is infinite, the procedure will only be approximate, because, generally, the contact domain will not be strictly elliptical in that case.

### 3.1. Macroscopic Contact Solution

The contact solution is based on the integral expression (6) for the force as a function of the indentation depth and the contact domain. For profiles in the form of (15), the contact domain is elliptical, and so *p** and the contour of the contact domain are provided by Equations (11) and (12). Hence, the normal force integral reads
(16)$$F=\int_{0}^{2\pi }\int_{0}^{R\left(\varphi \right)}\left[d-B_{n}r^{2n}{\left(1-\varepsilon^{2}\cos^{2}\varphi \right)}^{n}\right]\frac{c}{2b^{1-k}\;K_{k}\left(e\right)}{\left(1-\frac{r^{2}}{R{\left(\varphi \right)}^{2}}\right)}^{\frac{k-1}{2}}\;rdrd\varphi ,$$
which can be evaluated as
(17)$$\begin{array}{c} \;\;\;\;F=\frac{\pi c}{K_{k}\left(e\right)\sqrt{1-e^{2}}}\left[\frac{db^{1+k}}{1+k}-B_{n}\frac{\kappa \left(2n,k\right)}{2n+k+1}b^{2n+k+1}\frac{P_{n}\left(e^{2},\varepsilon^{2}\right)}{{\left(1-e^{2}\right)}^{n}}\right],\\ \;\kappa \left(m,k\right)=\frac{\Gamma \left(\left(1+k\right)/2\right)\Gamma \left(1+m/2\right)}{\Gamma \left(\left(k+m+1\right)/2\right)},\;\;\;\\ \;P_{n}\left(e^{2},\varepsilon^{2}\right)=\frac{{\left(1-e^{2}\right)}^{n+1/2}}{2\pi }\int_{0}^{2\pi }\frac{{\left(1-\varepsilon^{2}\cos^{2}\varphi \right)}^{n}d\varphi }{{\left(1-e^{2}\cos^{2}\varphi \right)}^{n+1}}. \end{array}$$ Note that *P_n_* are polynomials of the order *n* in *e*^2^ and *ε*^2^, which have no dependence on *k*. In other words, they are already known from the corresponding homogeneous solution.

Let us characterize the contact domain by the smaller half-axis *b* and the eccentricity *e*. According to Barber’s theorem, the values of *b* and *e*, which solve the non-adhesive normal contact problem, will maximize the force in Equation (17) for a given indentation depth. Maximizing with respect to *b*, we obtain
(18)$$\frac{\partial F}{\partial b}=0\;\;\;\;\Rightarrow \;\;\;\;d=B_{n}\kappa \left(2n,k\right)b^{2n}\frac{P_{n}\left(e^{2},\varepsilon^{2}\right)}{{\left(1-e^{2}\right)}^{n}}.$$

Hence,
(19)$$F=\frac{2\pi nc}{K_{k}\left(e\right)\left(2n+k+1\right)\left(1+k\right)}d^{\frac{2n+k+1}{2n}}\sqrt{{\left(1-e^{2}\right)}^{k}}{\left[B_{n}\kappa \left(2n,k\right)P_{n}\left(e^{2},\varepsilon^{2}\right)\right]}^{-\frac{1+k}{2n}},$$
and maximizing with respect to *e* leads to the condition
(20)$$1-\frac{D_{k}\left(e\right)}{K_{k}\left(e\right)}=\frac{e}{1-k}\left[\frac{1+k}{2nP_{n}\left(e^{2},\varepsilon^{2}\right)}\frac{\partial P_{n}}{\partial e}+\frac{ke}{1-e^{2}}\right],\;\;D_{k}\left(e\right)=\int_{0}^{\frac{\pi }{2}}{\left(1-e^{2}\cos^{2}\varphi \right)}^{\frac{k-3}{2}}d\varphi .$$

Equations (18)–(20) exactly solve the normal contact problem. For *e* = *ε* = 0, it is easily verified that the known axisymmetric results [7] (p. 265) are exactly recovered. Note that the trivial solution of Equation (20), *e* = 0, corresponds to a minimum of the force. For *k* = 0, the results simplify to the known homogeneous solution.

### 3.2. Pressure Distribution

For the determination of the pressure distribution, we return to the idea of understanding the indentation process as a series of incremental flat punch indentations.

As Equation (20) has no dependence on the normal load (i.e., either the indentation depth or the normal force), the contact eccentricity will remain constant during the indentation process. Moreover, the pressure distribution at the end of the indentation procedure directly follows from the integral of pressure distributions for the incremental flat punch indentations. Hence, in polar coordinates, using Equations (11), (12), and (18), we obtain
(21)$$\begin{array}{cc} p\left(r,\varphi \right) & =\frac{c}{2K_{k}\left(e\right)}\int_{r}^{R\left(\varphi \right)}{\left[b\left(R^{*}\left(\varphi \right)\right)\right]}^{k-1}\frac{R^{*}{\left(\varphi \right)}^{1-k}}{\sqrt{{\left(R^{*}{\left(\varphi \right)}^{2}-r^{2}\right)}^{1-k}}}\frac{dd^{*}}{db^{*}}\frac{db^{*}}{dR^{*}\left(\varphi \right)}dR^{*}\left(\varphi \right)\\ & =\frac{ncd}{b^{1-k}K_{k}\left(e\right)}\int_{\rho }^{1}\frac{u^{2n-1}du}{\sqrt{{\left(u^{2}-\rho^{2}\right)}^{1-k}}},\;\;\;\;\rho^{2}:=\frac{r^{2}}{R{\left(\varphi \right)}^{2}}=\frac{x^{2}}{a^{2}}+\frac{y^{2}}{b^{2}},\;\;\;\;\rho \leq 1. \end{array}$$
It is interesting that, except for the constant factor before the integral, this corresponds to the known axisymmetric solution [7] (p. 265) scaled to the elliptical contact domain.

### 3.3. Example: Elliptical Hertzian Contact

Based on the general solution for arbitrary (but natural) *n*, let us recover the known solution for the Hertzian case with *n* = 1. We have *P*_1_(*e*^2^, *ε*^2^) = (2 − *e*^2^ − *ε*^2^)/2 and, therefore,
(22)$$\begin{array}{c} d=\frac{\left(2-e^{2}-\varepsilon^{2}\right)B_{1}b^{2}}{\left(1+k\right)\left(1-e^{2}\right)},\\ F=\frac{2\pi c\left(2-e^{2}-\varepsilon^{2}\right)B_{1}b^{3+k}}{K_{k}\left(e\right){\left(1-e^{2}\right)}^{3/2}{\left(1+k\right)}^{2}\left(3+k\right)}. \end{array}$$

The contact eccentricity is defined as the nontrivial solution of
(23)$$\frac{D_{k}\left(e\right)}{K_{k}\left(e\right)}-1=\frac{e^{2}}{1-k}\frac{1}{2-e^{2}-\varepsilon^{2}}\left(1-k\frac{1-\varepsilon^{2}}{1-e^{2}}\right),$$
which can be shown to coincide with the known result [23].

In the limiting case *k* → 1, the solution is always *e* = *ε*. This is to be expected, as the Gibson solid (with *k* = 1 and *ν* = 0.5) reacts like a three-dimensional foundation of independent linear springs and, therefore, the contact domain always corresponds to horizontal indenter cross sections. For small eccentricities, Equation (23) has the asymptotic solution
(24)$$e\approx \sqrt{\frac{4}{3+k}}\;\varepsilon .$$

In Figure 1, the relation between the contact eccentricity *e* and the profile eccentricity *ε* is shown for three values of *k*, based on the exact solution (23). The thin drawn-through lines correspond to the respective asymptotic solution (24), which apparently is valid up to *ε* ≈ 0.2.

Obviously, for increasing values of *k*, the contact eccentricity becomes smaller, i.e., closer to the profile eccentricity. This is in line with the above comment that, for *k* → 1, the solution is always *e* = *ε*. Finally, the pressure distribution equals
(25)$$p\left(x,y\right)=\frac{cd}{b^{1-k}\left(1+k\right)K_{k}\left(e\right)}{\left(1-\frac{x^{2}}{a^{2}}-\frac{y^{2}}{b^{2}}\right)}^{\frac{1+k}{2}},\;\;\;\;\frac{x^{2}}{a^{2}}+\frac{y^{2}}{b^{2}}\leq 1,\;\;\;\;a^{2}=\frac{b^{2}}{1-e^{2}}.$$

## 4. A Fabrikant-Type Approximation for General Contact Domains

To constructively apply the extremal principle (5) to the force integral (6), one requires an expression for the pressure distribution *p** under a flat punch with arbitrary cross section. In that regard, the form (11) for the pressure distribution under a rigid flat elliptical punch suggests that Fabrikant’s approximation [24] for the pressure distribution under a flat punch of arbitrary planform, indenting a homogeneous elastic half-space, can be generalized straightforwardly to account for the type of functional elastic inhomogeneity considered in the present manuscript.

Noteworthily—as was pointed out very recently in [33]—what generally is referred to as “Fabrikant’s approximation” is, in fact, a special case of an asymptotic expansion, which was developed earlier by the group of Mossakovskii [34]. Nevertheless, to avoid confusion, we will keep to the usual term.

Let us suppose a flat punch of arbitrary shape with the contour *R*(*φ*) in polar coordinates is pressed into a power-law graded elastic half-space. We postulate that the pressure distribution will be defined approximately by the expression (11) (which is, of course, exact for the elliptical flat punch). Accordingly, the total normal force will be
(26)$$F=\int_{0}^{2\pi }\int_{0}^{R\left(r\right)}p\left(r,\varphi \right)rdrd\varphi \approx \frac{2p_{0}A}{1+k},\;\;\;\;A=\frac{1}{2}\int_{0}^{2\pi }R{\left(\varphi \right)}^{2}d\varphi ,$$
and, similarly, for the indentation depth,
(27)$$d=w\left(x=0,y=0\right)\approx \frac{p_{0}J_{k}}{2c},\;\;\;\;J_{k}=\int_{0}^{2\pi }R{\left(\varphi \right)}^{1-k}d\varphi .$$

Hence, the contact stiffness approximately equals
(28)$$K=\frac{F}{\delta }\approx \frac{2c}{1+k}\frac{\langle R^{2}\rangle }{\langle R^{1-k}\rangle },$$
where the brackets denote averaging over the polar angle.

### Example: Flat Punch with Square Planform

As an illustrative example, let us consider a square with side length 2*a* as the face of the flat punch. For *k* ≠ 0, we obtain for the contact stiffness
(29)$$K\approx \frac{4ca^{1+k}}{1+k}{\left[B\left(1;\frac{k}{2},\frac{1}{2}\right)-B\left(\frac{1}{2};\frac{k}{2},\frac{1}{2}\right)\right]}^{-1},$$
with the incomplete Beta function B. Hence, if we write *K* = *ca*^1+*k*^/(1 + *k*)/*S*, we obtain for the shape factor *S*
(30)$$S_{F}\approx \frac{1}{4}\left[B\left(1;\frac{k}{2},\frac{1}{2}\right)-B\left(\frac{1}{2};\frac{k}{2},\frac{1}{2}\right)\right].$$

It is interesting to compare this value with the result of Boyer’s [35] approximate solution procedure for the same problem. Partitioning the square into four sub-squares with side length *a*, and calculating the displacement of one sub-square under the assumption that the loads on the other three sub-squares can be treated as point loads, applied at the sub-square center, one obtains with the fundamental solution (2) and above definition for the shape factor that
(31)$$S_{B}\approx \frac{\cos\left(k\pi /2\right)}{\pi \left(1+k\right)}\frac{1+\sqrt{2^{-\left(3+k\right)}}}{2-2^{k}}.$$

It should be noted that the term “shape factor” in this framework can be slightly misleading, as the power-law graded elastic half-space has an intrinsic length scale *z*_0_, and therefore any contact problem of power-law graded materials also has size effects; this is blurred in the above definition (which is chosen such that the shape factor for the cylindrical flat punch always equals 1/2).

In Figure 2, the shape factors according to the approximations (30) and (31) are shown in comparison with a rigorous numerical solution obtained with the boundary element method (BEM) [36]. While Fabrikant’s approximation for all *k* agrees very closely with the numerical solution, Boyer’s approximation only in the homogeneous case provides satisfactory results.

Nevertheless, the “corner singularity” of the pressure distribution at the corners of the square is, in fact, more severe than predicted by Fabrikant’s approximation, which only obtains an “edge stress singularity” for every point of the contact boundary. In other words, it treats all boundary points as parts of an edge, which is smooth along the contact contour. This effect is demonstrated in Figure 3. There, the BEM results for the pressure distributions along the *x*-axis and along the diagonal of the contact square are shown in normalized variables for a graded elastic half-space with *k* = 0.5 and compared with the respective results from the generalization of Fabrikant’s approximation. While very good agreement between the approximation and the rigorous numerical solution is achieved along the *x*-axis, the approximation underestimates the pressure distribution along the diagonal and, especially, the stress singularity at the corner. Thus, a more detailed asymptotic analysis of the corner singularity behavior will probably result in a more refined approximation for the pressure distribution.

## 5. General Approximate Solution for Almost Axisymmetric Contacts

Nevertheless, not only contact problems of flat punches with arbitrary contact domains can be solved approximately for homogeneous materials, but also general smooth profiles have been considered. Recently, based on Fabrikant’s approximation and Barber’s extremal principle, Popov [25] presented an analytic (first order) approximate contact solution for the indentation of a homogeneous elastic half-space by a rigid indenter, whose shape slightly deviates from axis-symmetry but is otherwise arbitrary. As now all constituent parts of that solution have provided for the case of the power-law graded elastic half-space, we can easily generalize Popov’s solution for the elastically inhomogeneous problem.

### 5.1. General Approximate Solution Procedure for Non-Symmetric Profiles

The solution is based on a general approximate procedure, which, for the homogeneous case, was first suggested by Barber and Billings [37] and can straightforwardly be generalized for the power-law graded problem.

Let us suppose a rigid indenter with the profile *f* = *f*(*r*,*φ*) is pressed into the power-law graded elastic half-space. There shall be no tilting moments around the origin of the coordinate system, defined such that *f*(*r* = 0) = 0. The normal force is defined by Equation (6). If, for the pressure distribution *p**, we use the generalization of Fabrikant’s approximation, discussed in the previous section, the force integral approximately simplifies to
(32)$$F\approx \frac{2c}{J_{k}}\int_{0}^{2\pi }\frac{d}{1+k}R{\left(\varphi \right)}^{2}-R{\left(\varphi \right)}^{1-k}G\left(R\left(\varphi \right),\varphi \right)d\varphi ,\;\;\;\;J_{k}=\int_{0}^{2\pi }R{\left(\varphi \right)}^{1-k}d\varphi ,$$
with the generalized Abel transform
(33)$$G\left(R\left(\varphi \right),\varphi \right)=\int_{0}^{R\left(\varphi \right)}\frac{f\left(r,\varphi \right)r\;dr}{\sqrt{{\left(R\left(\varphi \right)-r^{2}\right)}^{1-k}}},\;\;\;\;g\left(R\left(\varphi \right),\varphi \right)=\frac{\partial G\left(R\left(\varphi \right),\varphi \right)}{\partial R\left(\varphi \right)}.$$

According to Barber’s principle, applied to the power-law graded case, the correct contour *R*(*φ*) of the contact domain for a provided indentation depth results from the solution of the variational problem (5). Let us first find the correct contour for fixed values of the non-linear moment *J_k_*. According to Lagrange, this is carried out by finding the unconditional extremum of the functional
(34)$$L=\frac{2c}{J}\int_{0}^{2\pi }\frac{d}{1+k}R{\left(\varphi \right)}^{2}-R{\left(\varphi \right)}^{1-k}G\left(R\left(\varphi \right),\varphi \right)d\varphi -\lambda \left(\int_{0}^{2\pi }R{\left(\varphi \right)}^{1-k}d\varphi -J_{k}\right),$$
with the Lagrange multiplier *λ*. Applying the extremal principle results in
(35)$$\frac{2c}{J_{k}}\left[\frac{2d}{1+k}R{\left(\varphi \right)}^{1+k}-\left(1-k\right)G\left(R\left(\varphi \right),\varphi \right)-R\left(\varphi \right)g\left(R\left(\varphi \right),\varphi \right)\right]=\lambda \left(1-k\right).$$

In general, this will be a transcendent algebraic equation for *R*(*φ*), defying a closed-form solution. However, in analogy to the homogeneous case, a closed-form asymptotic solution can be derived if the profile is only slightly deviating from axial symmetry.

### 5.2. Closed-Form Approximate Solution for Almost Axisymmetric Profiles

Considering the almost axially symmetric problem, we write
(36)$$f\left(r,\varphi \right)=f_{0}\left(r\right)+f_{1}\left(r,\varphi \right),\;\;\;\;\forall \left\{r,\varphi \right\}:f_{1}\left(r,\varphi \right)\ll f_{0}\left(r\right)=\langle f\left(r,\varphi \right)\rangle ,$$
and also introduce the corresponding generalized Abel transforms
(37)$$\begin{array}{c} G_{0}\left(R\left(\varphi \right)\right)=\int_{0}^{R\left(\varphi \right)}\frac{f_{0}\left(r\right)r\;dr}{\sqrt{{\left(R\left(\varphi \right)-r^{2}\right)}^{1-k}}},\;\;\;\;g_{0}\left(R\left(\varphi \right)\right)=\frac{dG_{0}\left(R\left(\varphi \right)\right)}{dR\left(\varphi \right)},\\ G_{1}\left(R\left(\varphi \right),\varphi \right)=\int_{0}^{R\left(\varphi \right)}\frac{f_{1}\left(r,\varphi \right)r\;dr}{\sqrt{{\left(R\left(\varphi \right)-r^{2}\right)}^{1-k}}},\;\;\;\;g_{1}\left(R\left(\varphi \right),\varphi \right)=\frac{\partial G_{1}\left(R\left(\varphi \right),\varphi \right)}{\partial R\left(\varphi \right)}. \end{array}$$

The contact domain will be almost circular with the contour
(38)$$R\left(\varphi \right)=R_{0}+R_{1}\left(\varphi \right),\;\;\;\;\forall \varphi :R_{1}\left(\varphi \right)\ll R_{0}.$$

Substituting Equation (38) into Equation (35), expanding in powers of *R*_1_/*R*_0_, and neglecting all terms of higher than linear order, we obtain
(39)$$R_{1}\left(\varphi \right)\approx \frac{\frac{\pi \lambda \left(1-k\right)R_{0}^{1-k}}{c}-\frac{2dR_{0}^{1+k}}{1+k}+\left(1-k\right)G\left(R_{0},\varphi \right)+R_{0}g\left(R_{0},\varphi \right)}{2dR_{0}^{k}-\left(2-k\right)g\left(R_{0},\varphi \right)-R_{0}g^{\prime }\left(R_{0},\varphi \right)},$$
where the prime denotes the derivative with respect to the first functional argument.

The determination of the Lagrange multiplier is completely analogous to the homogeneous derivation presented by Popov [25] and shall therefore not be detailed here for reasons of space. We obtain, after some elementary calculations,
(40)$$R_{1}\left(\varphi \right)\approx \frac{\left(1-k\right)G_{1}\left(R_{0},\varphi \right)+R_{0}g_{1}\left(R_{0},\varphi \right)}{2dR_{0}^{k}-\left(2-k\right)g_{0}\left(R_{0}\right)-R_{0}{g^{\prime }}_{0}\left(R_{0}\right)}.$$

Hence, the force integral (32) further simplifies to
(41)$$\begin{array}{cc} F & \approx \frac{c}{\pi R_{0}^{1-k}}\{\int_{0}^{2\pi }\frac{dR_{0}^{2}}{1+k}-R_{0}^{1-k}G_{0}\left(R_{0}\right)d\varphi -\int_{0}^{2\pi }R_{0}^{1-k}G_{1}\left(R_{0},\varphi \right)d\varphi +\\ & +\int_{0}^{2\pi }\left[\frac{2dR_{0}}{1+k}-\left(1-k\right)R_{0}^{-k}G_{0}\left(R_{0}\right)-R_{0}^{1-k}g_{0}\left(R_{0}\right)\right]R_{1}\left(\varphi \right)d\varphi \}. \end{array}$$

Here, the second and third integrals vanish in perfect analogy to the respective homogeneous solution (see [25] for the respective details). Finding the extremum of the remaining trivial first integral with respect to *R*_0_ results in the axisymmetric solutions
(42)$$d\approx R_{0}^{-k}g_{0}\left(R_{0}\right),$$
and
(43)$$F\approx 2c\left[\frac{R_{0}g_{0}\left(R_{0}\right)}{1+k}-G_{0}\left(R_{0}\right)\right].$$

Finally, with Equation (42), Equation (40) can be further simplified to provide
(44)$$R\left(\varphi \right)\approx R_{0}+\frac{\left(1-k\right)G_{1}\left(R_{0},\varphi \right)+R_{0}g_{1}\left(R_{0},\varphi \right)}{kg_{0}\left(R_{0}\right)-R_{0}{g^{\prime }}_{0}\left(R_{0}\right)}.$$

Equations (42)–(44) define the closed-form analytic, approximate solution to the almost axially symmetric Boussinesq problem. Of course, for *k* = 0, the known homogeneous approximate solution is exactly recovered. Note that either *d*, *F*, or *R*_0_ can be used to completely characterize the contact configuration.

### 5.3. Pressure Distribution for Almost Axisymmetric Profiles

As was pointed out before, the indentation process from the indentation depth *d** = 0 to *d** = *d* can be understood as a series of incremental flat punch indentations d*d**, with the planform of the punch coinciding with the contact domain at the current indentation depth *d**. Once the macroscopic contact problem has been solved, i.e., the contact domain as a function of the indentation depth has been determined, the resulting pressure distribution can be obtained as the integral of the pressure distributions from the corresponding series of incremental flat punch indentations. In an approximate sense, this procedure can always be executed analytically based on the approximation for the pressure distribution under a flat punch of arbitrary shape. Hence, we obtain for the pressure distribution
(45)$$p\left(r,\varphi \right)\approx \frac{c}{\pi }\int_{r}^{R\left(\varphi \right)}{\left(R_{0}^{*}\right)}^{k-1}\frac{R^{*}{\left(\varphi \right)}^{1-k}}{\sqrt{{\left(R^{*}{\left(\varphi \right)}^{2}-r^{2}\right)}^{1-k}}}\frac{dd^{*}}{dR_{0}^{*}}\frac{\partial R_{0}^{*}}{\partial R^{*}\left(\varphi \right)}dR^{*}\left(\varphi \right),$$
where *R*_0_ must be understood as a function of *R*(*φ*), by inverting Equation (44).

### 5.4. Example I: Elliptical Hertzian Contact

As an illustrative example, let us consider the elliptical Hertzian contact with
(46)$$f\left(r,\varphi \right)=r^{2}\left(A\cos^{2}\varphi +B\sin^{2}\varphi \right),\;\;\;\;A<B.$$

The separation into axisymmetric and non-axisymmetric parts results in
(47)$$f_{0}\left(r\right)=r^{2}\frac{A+B}{2},\;\;\;\;f_{1}\left(r,\varphi \right)=r^{2}\frac{A-B}{2}\cos\left(2\varphi \right).$$

Hence,
(48)$$\begin{array}{c} G_{0}\left(R_{0}\right)=\frac{R_{0}^{3+k}\left(A+B\right)}{\left(3+k\right)\left(1+k\right)},\;\;\;\;g_{0}\left(R_{0}\right)=\frac{R_{0}^{2+k}\left(A+B\right)}{1+k},\\ G_{1}\left(R_{0},\varphi \right)=\frac{R_{0}^{3+k}\left(A-B\right)}{\left(3+k\right)\left(1+k\right)}\cos\left(2\varphi \right),\;\;\;\;g_{1}\left(R_{0},\varphi \right)=\frac{R_{0}^{2+k}\left(A-B\right)}{1+k}\cos\left(2\varphi \right), \end{array}$$
and Equations (42)–(44) provide the approximate contact solution
(49)$$d\approx \frac{R_{0}^{2}\left(A+B\right)}{1+k},\;\;\;\;F\approx \frac{4cR_{0}^{3+k}\left(A+B\right)}{{\left(1+k\right)}^{2}\left(3+k\right)},\;\;\;\;R\left(\varphi \right)\approx R_{0}\left[1-\frac{A-B}{A+B}\frac{2}{3+k}\cos\left(2\varphi \right)\right].$$

For small eccentricities, the last relation describes an ellipse with half-axes
(50)$$a\approx R_{0}\left[1+\frac{2}{3+k}\frac{B-A}{A+B}\right],\;\;\;\;b\approx R_{0}\left[1-\frac{2}{3+k}\frac{B-A}{A+B}\right],\;$$
and the eccentricity of the contact domain is approximately provided by
(51)$$e^{2}=1-\frac{b^{2}}{a^{2}}\approx \frac{8}{3+k}\frac{B-A}{A+B}=\frac{8}{3+k}\frac{\varepsilon^{2}}{2-\varepsilon^{2}}\approx \frac{4\varepsilon^{2}}{3+k},\;\;\;\;\varepsilon^{2}=1-\frac{A}{B},$$
which is in perfect agreement with Equation (24). It is also checked easily that, for small eccentricities, the obtained solutions for *d* and *F* agree with the asymptotes of the respective exact solutions in Equation (22).

### 5.5. Example II: Indentation by a Pyramid with Square Planform

As a second example for the approximate procedure, let us consider the normal contact with a rigid pyramid of square planform. The application of the analytic approximate solutions (42)–(44) for this profile geometry is straightforward and shall not be detailed here for reasons of space. In Figure 4, a comparison is shown between a BEM-based numerical solution and the approximate solution for the contour of the contact domain in normalized variables. As expected, the analytic approximate solution works very well, except for the edges of the pyramid, for which the contact domain is “smoothed” in the approximate solution.

## 6. Discussion and Conclusions

Several non-adhesive, frictionless indentation problems for the power-law graded elastic half-space have been considered in the case of elliptical and non-standard contact domains. Different powerful techniques, which were originally developed for the closed-form analysis of the Boussinesq problem for a homogenous half-space, and which already proved their usefulness, were generalized for the elastically inhomogeneous problem. Nonetheless, some restrictions have to be kept in mind when applying the obtained results to real engineering contacts.

These restrictions can be classified into two groups. The first stems from the physical modelling and mainly comprises the assumptions of linearly elastic material behavior and the very specific form of functional grading (i.e., the power law). Real contacts will usually exhibit different forms of inelasticity; moreover, the precise form of the functional elastic grading might deviate from the power-law description, especially if it originates from a hard-to-control technical grading procedure. However, the power law is a very convenient way of approximating more complicatedly graded properties, and, due to the applicability of all results for positive and negative values of the grading exponent, allows for the analysis of both hard surfaces with a softer material core as well as soft surfaces with a harder material core.

The second group of restrictions stems only from the mathematical analysis. As the Boussinesq problem for a general contact domain, despite its linearity, cannot be solved analytically in closed form, several approximate procedures were facilitated to allow for a fast, analytic solution of the contact problem. Although these procedures are asymptotically exact, i.e., they provide exact first-order solutions of a general problem around a specific expansion “point” (usually, the axisymmetric problem), their applicability should, if possible, be checked by more rigorous numerical simulations, e.g., based on the BEM or the finite element method (FEM), at least for some random problem samples.

On the other hand, the obtained normal contact solutions can be used to solve other classes of contact problems that can, exactly or approximately, be reduced to the elastic Boussinesq problem. This is true for viscoelastic normal contacts, via the elastic–viscoelastic correspondence principle [38], or tangential contacts with friction, via the generalization for power-law graded materials [19] of the principle by Jäger [39] and Ciavarella [40].

Finally, the approximate solution presented in Section 5 can be considered an extension of the method of dimensionality reduction (MDR) for power-law graded elastic materials [18] to non-axisymmetric contacts. It can be used for different problems in tribology and engineering, e.g., within the framework of indentation testing or as a tribological tool for the analysis of local features of a random rough surface.

## Figures and Tables

**Figure 1 materials-16-04364-f001:**
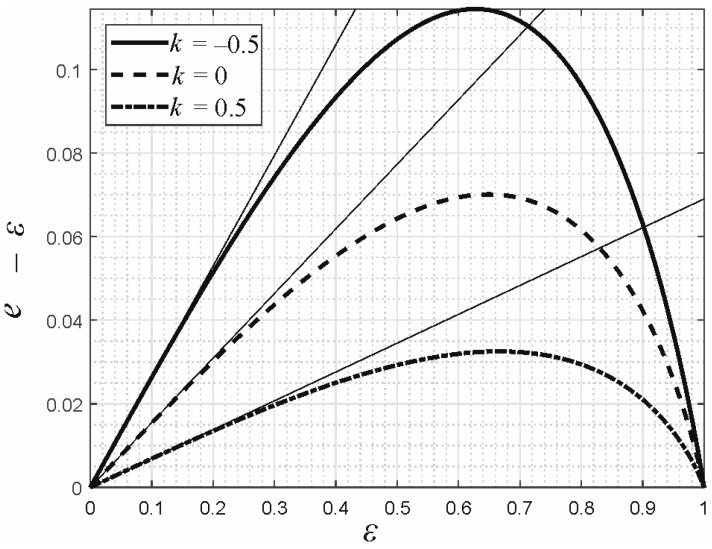
Difference between the eccentricities of the contact area and the indenter profile as a function of the profile eccentricity for the elliptical Hertzian contact and different exponents of the power-law grading. The thin lines correspond to the asymptotic solution (24).

**Figure 2 materials-16-04364-f002:**
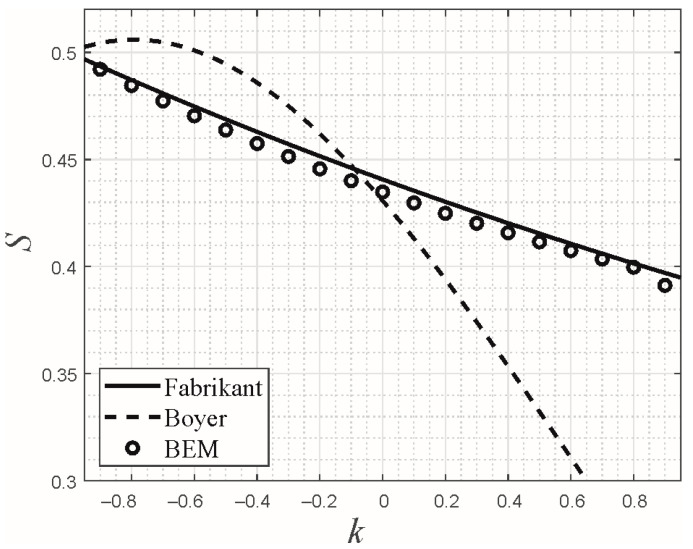
Comparison of the shape factor for a flat punch with square planform as a function of the exponent *k* of the power-law grading according to the (generalized) approximations by Fabrikant (30) and Boyer (31), with a rigorous numerical solution obtained with the boundary element method (BEM).

**Figure 3 materials-16-04364-f003:**
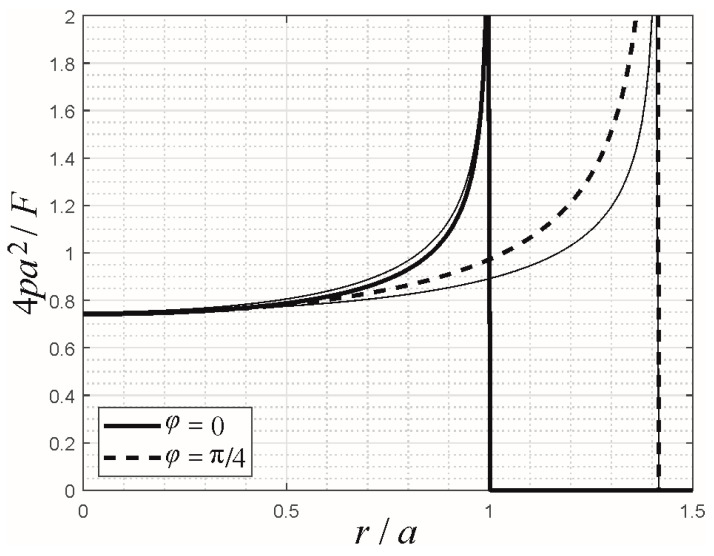
Pressure distributions (obtained by BEM) in normalized variables for the indentation of a power-law graded elastic half-space with *k* = 0.5 by a flat punch with square planform, along the *x*-axis (*φ* = 0) and along the diagonal (*φ* = π/4). The thin drawn-through lines denote the respective results from the generalization of Fabrikant’s approximation.

**Figure 4 materials-16-04364-f004:**
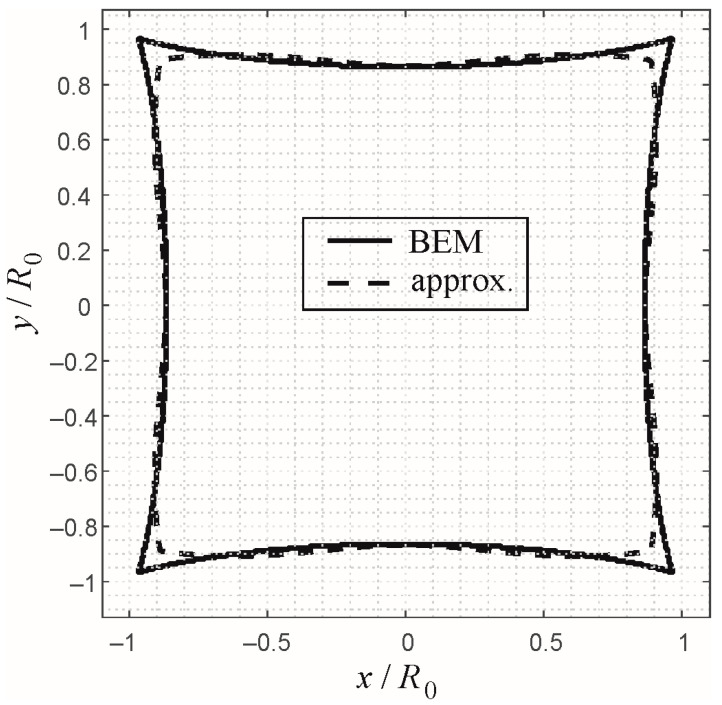
Boundary of the contact domain in normalized variables for the indentation of a power-law graded elastic half-space with exponent *k* = 0.5 by a rigid pyramid with square planform. Solid line: numerical solution based on BEM. Dashed line: approximate solution based on the analytic procedure for almost axisymmetric profiles.

## Data Availability

Not applicable.

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
