# Peer review of "On Boussinesq’s Problem for a Power-Law Graded Elastic Half-Space on Elliptical and General Contact Domains"

_materials, 2023, doi:10.3390/ma16124364_

Round 1
Reviewer 1 Report
In this paper the author employ Boundary Element Method (BEM) and analytical solution for investigating Boussinesq’s Problem for a Power-Law Graded Elastic Half-Space on Elliptical and General Contact Domains. The topic is interesting and the paper is well written and organized. Based on reviewer knowledge, this paper has positive impact on scientific society. I strongly recommend it for publishing after minor revision as the following:
1) The difference between present work and previous studied should be more emphasized in the introduction. (Novelty aspect)
2) In figure 4, it is better to explain the solid line and dashed line inside the figure
3) Eq .1 is needed the reference. Please add or some references for this relation.
4) Can author add the physical reason for the numerical, analytical results?
5) Please polish the manuscript carefully.
6) There are a lot of important research about FGM. Please consider more references about FGM in the literature. For instance:
Free vibration analysis of FG porous joined truncated conical-cylindrical shell reinforced by graphene platelets (journal: Advances in nano research )
Author Response
Please see the attached answer file.

Author Response
Please see the attached answer file.

Round 2
